# Test-time augmentation with uncertainty estimation for deep learning-based medical image segmentation

**Guotai Wang**[*]
University College London
guotai.wang.14@ucl.ac.uk

**Wenqi Li**[*]
University College London
wenqi.li@ucl.ac.uk

**Michael Aertsen**[†]
KU Leuven
michael.aertsen@uzleuven.be

**Jan Deprest**[†]
KU Leuven
jan.deprest@uzleuven.be

**Sébastien Ourselin**[*]
University College London
s.ourselin@ucl.ac.uk

**Tom Vercauteren**[*]
University College London
t.vercauteren@ucl.ac.uk

## Abstract

Data augmentation has been widely used for training deep learning systems for medical image segmentation and plays an important role in obtaining robust and transformation-invariant predictions. However, it has seldom been used at test time for segmentation and not been formulated in a consistent mathematical framework. In this paper, we first propose a theoretical formulation of test-time augmentation for deep learning in image recognition, where the prediction is obtained through estimating its expectation by Monte Carlo simulation with prior distributions of parameters in an image acquisition model that involves image transformations and noise. We then propose a novel uncertainty estimation method based on the formulated test-time augmentation. Experiments with segmentation of fetal brains and brain tumors from 2D and 3D Magnetic Resonance Images (MRI) showed that 1) our test-time augmentation outperforms a single-prediction baseline and dropout-based multiple predictions, and 2) it provides a better uncertainty estimation than calculating the model-based uncertainty alone and helps to reduce overconfident incorrect predictions.

## 1 Introduction

In recent years, deep learning has become the state-of-the-art method for medical image recognition tasks such as image classification, object detection and segmentation [8]. As a type of data-driven approach, it learns features automatically, without explicitly modeling the complex variations of images. From the perspective of image acquisition, the acquired image may contain noise related to the environment, and different viewpoints can lead to transformed versions of the same object. It is desirable to enable a recognition system to be robust against noise and transformations, leading to noise-invariant and transformation-invariant recognition, e.g., rotation-, translation- and scale-invariant. Convolutional Neural Networks (CNN) are designed to be translation-invariant by sharing

---

[*]Department of Medical Physics and Biomedical Engineering, Wellcome EPSRC Centre for Interventional and Surgical Sciences (WEISS), University College London, London, WC1E 6BT, UK. Tom Vercauteren is also with KU Leuven.

[†]Department of Obstetrics & Gynaecology, and Radiology, University Hospitals KU Leuven, 3000 Leuven, Belgium. Jan Deprest is also with WEISS and Institute for Women's Health, University College London.

1st Conference on Medical Imaging with Deep Learning (MIDL 2018), Amsterdam, The Netherlands.

weights at different positions of the image. Dropout techniques [2] have been used to make the model more robust to noise. However, CNNs are not inherently invariant to more general transformations.

To alleviate this problem, many researchers have tried to collect a training dataset that is as large as possible in order to include a large variation of image contexts to train the model. When collecting such a large dataset is difficult or impossible, data augmentation is commonly used to enlarge a relatively small dataset by applying transformations to its samples to create new ones for training [6], and this helps to improve invariance to spatial transformations at test time. The transformations for augmentation typically include flipping, cropping, rotating, and scaling training images. Krizhevsky et al. [6] also altered the intensities of the training images for data augmentation. In [1, 13], elastic deformations were used for biomedical image segmentation. Recently, convex combination of pairs of samples has been proposed as a way of data augmentation for training [18].

While data augmentation is typically employed for training of CNNs, using it at test time has seldom been investigated. Only few studies have empirically found that combining predictions of multiple transformed versions of a test image helps to improve the performance. For example, Matsunaga et al. [10] geometrically transformed test images for skin lesion classification. Radosavovic et al. [12] used a single model to predict multiple transformed copies of unlabeled images for data distillation. Jin et al. [4] tested on samples extended by rotation and translation for pulmonary nodule detection. However, all these methods used data augmentation for testing as an ad hoc method, without detailed formulation or theoretical explanation.

In addition to robustness to imaging conditions, uncertainty estimation plays a critical role in medical image recognition. For example, for chest X-ray image classification [16], a testing result with high uncertainty may need a human expert to give a decision. In a segmentation task, the predicted labels of pixels near the boundary of organs are likely to be uncertain [15], which can be used to guide user interactions. Several methods have been proposed for the estimation of model uncertainty. Exact Bayesian models offer a mathematically grounded method to infer model uncertainty, but they are hard to implement for CNNs. Alternatively, it has been shown that dropout can be cast as a Bayesian approximation to represent model uncertainty [2]. In [19], Stein Variational Gradient Descent (SVGD) was used to perform approximate Bayesian inference on uncertain CNN parameters. In [7], ensembles of multiple models were proposed for uncertainty estimation. However, these methods only consider the uncertainty resulting from the trained models, and tend to produce overconfident incorrect predictions. Kendall et al. [5] proposed a framework based on Bayesian deep learning to model uncertainties related not only to network parameters but also to image noise. In addition to the model and image noise, the prediction uncertainty of the observed image may also be related to viewpoints or transformations of the inherent object, as such factors also affect the prediction output, leading to more uncertainty that depends on the input. To the best of our knowledge, the uncertainty related to viewpoints or transformations has rarely been investigated for deep learning with CNNs.

*Summary of contributions*: Our contribution in this paper is two-fold. First, we propose a theoretical formulation of test-time augmentation for deep learning. We represent an image as a result of an acquisition process which involves geometric transformations and image noise. We model the hidden parameters of the image acquisition process with prior distributions, and infer the prediction output for a given image by estimating its expectation with a Monte Carlo simulation process. The formulation is a mathematical explanation of test-time augmentation that is general for image recognition tasks. Second, we propose a novel uncertainty estimation method based on the formulated test-time augmentation for image recognition tasks. We demonstrate the effect of test-time augmentation with 2D and 3D segmentation tasks, and show that our proposed method provides a better uncertainty estimation with fewer overconfident incorrect predictions than using model-based uncertainty.

## 2   Methods

The proposed method with test-time augmentation includes two parts. The first part is a mathematical representation of ensembles of predictions of multiple transformed versions of the input. In Section 2.1, we represent an image as a result of an image acquisition model with hidden parameters. In Section 2.2, we formulate test-time augmentation as inference with hidden parameters following given prior distributions. The second part calculates the diversity of the prediction results of an augmented test image, and it is used for estimation of the uncertainty related to image transformations and noise. This will be detailed in Section 2.3.

## 2.1 Image acquisition model

The image acquisition model describes the process by which the observed images have been obtained. This process is confronted with a lot of factors that can be related or unrelated to the imaged object, such as blurring, down-sampling, spatial transformation, and system noise. While blurring and down-sampling are commonly considered for image super-resolution [17], in the context of image recognition they have a relatively lower impact. Therefore, we focus on the spatial transformation and noise, with adding intensity changes being a straightforward extension.

$$X = \mathcal{T}_{\boldsymbol{\beta}}(X_0) + \boldsymbol{e} \tag{1}$$

where $X_0$ is an underlying image in a different position and orientation, i.e., a hidden variable. $\mathcal{T}$ is a transformation operator that is applied to $X_0$. $\boldsymbol{\beta}$ is the set of parameters of the transformation, and $\boldsymbol{e}$ represents the noise that is added to the transformed image. $X$ denotes the observed image that is used for inference at test time. Though the transformations can be in spatial, intensity or feature space, in this work we only study the impact of reversible spatial transformations (e.g., flipping, scaling, rotation and translation), which are the most common types of transformations occurring during image acquisition and used for data augmentation purposes. Let $\mathcal{T}_{\boldsymbol{\beta}}^{-1}$ denote the inverse transformation of $\mathcal{T}_{\boldsymbol{\beta}}$, then we have:

$$X_0 = \mathcal{T}_{\boldsymbol{\beta}}^{-1}(X - \boldsymbol{e}) \tag{2}$$

Similarly to data augmentation, we assume that $X$ and $X_0$ follow the same distribution. In a given application, this assumption leads to some prior distributions of the transformation parameters and noise. For example, in a 2D slice of fetal brain Magnetic Resonance Images (MRI), the orientation of the fetal brain can range among all the possible directions in a 2D plane, therefore the rotation angle $\boldsymbol{r}$ can be modeled with a uniform prior distribution $\boldsymbol{r} \sim U(0, 2\pi)$. The image noise is commonly modeled as a Gaussian distribution, i.e., $\boldsymbol{e} \sim \mathcal{N}(\boldsymbol{\mu}, \boldsymbol{\sigma})$, where $\boldsymbol{\mu}$ and $\boldsymbol{\sigma}$ are the mean and standard deviation respectively. Let $P_{\boldsymbol{\beta}}$ and $P_{\boldsymbol{e}}$ represent the prior distribution of $\boldsymbol{\beta}$ and $\boldsymbol{e}$ respectively, therefore we have $\boldsymbol{\beta} \sim P_{\boldsymbol{\beta}}$ and $\boldsymbol{e} \sim P_{\boldsymbol{e}}$.

Let $Y$ and $Y_0$ be the labels related to $X$ and $X_0$ respectively. For image classification, $Y$ and $Y_0$ are categorical variables, and they should be invariant with regard to transformations and noise, therefore $Y = Y_0$. For image segmentation, $Y$ and $Y_0$ are discretized label maps, and they are covariant with the spatial transformation, i.e., $Y = \mathcal{T}_{\boldsymbol{\beta}}(Y_0)$.

## 2.2 Inference with hidden variables

In the context of deep learning, let $f(\cdot)$ be the function represented by a neural network, and $\boldsymbol{\theta}$ represent the parameters learned from a set of training images with their corresponding annotations. In a standard formulation, the label $Y$ of a test image $X$ is inferred by:

$$Y = f(\boldsymbol{\theta}, X) \tag{3}$$

Since $X$ is only one of many possible observations of the underlying image $X_0$, direct inference with $X$ may lead to a biased result affected by the specific transformation and noise associated with $X$. To address this problem, we infer with the help of $X_0$ instead:

$$Y = \mathcal{T}_{\boldsymbol{\beta}}(Y_0) = \mathcal{T}_{\boldsymbol{\beta}}(f(\boldsymbol{\theta}, X_0)) = \mathcal{T}_{\boldsymbol{\beta}}\Big(f\big(\boldsymbol{\theta}, \mathcal{T}_{\boldsymbol{\beta}}^{-1}(X - \boldsymbol{e})\big)\Big) \tag{4}$$

where the exact values of $\boldsymbol{\beta}$ and $\boldsymbol{e}$ for $X$ are unknown. Instead of finding a deterministic prediction of $X$, we alternatively compute the distribution of $Y$ considering the distributions of $\boldsymbol{\beta}$ and $\boldsymbol{e}$.

$$P(Y) = P\left(\mathcal{T}_{\boldsymbol{\beta}}\Big(f\big(\boldsymbol{\theta}, \mathcal{T}_{\boldsymbol{\beta}}^{-1}(X - \boldsymbol{e})\big)\Big)\right), \text{where } \boldsymbol{\beta} \sim P_{\boldsymbol{\beta}}, \boldsymbol{e} \sim P_{\boldsymbol{e}} \tag{5}$$

To obtain the final prediction for $X$, we calculate the expectation of $Y$ using the distribution $P(Y)$.

$$E(Y) = \int yP(y)dy = \int_{\boldsymbol{\beta} \sim P_{\boldsymbol{\beta}}, \boldsymbol{e} \sim P_{\boldsymbol{e}}} \mathcal{T}_{\boldsymbol{\beta}}\Big(f\big(\boldsymbol{\theta}, \mathcal{T}_{\boldsymbol{\beta}}^{-1}(X - \boldsymbol{e})\big)\Big) P(\boldsymbol{\beta})P(\boldsymbol{e})d\boldsymbol{\beta}d\boldsymbol{e} \tag{6}$$

Calculating $E(Y)$ with Eq. (6) is computationally expensive, as $\boldsymbol{\beta}$ and $\boldsymbol{e}$ may take continuous values and $P_{\boldsymbol{\beta}}$ is a complex joint distribution of different types of transformations. Alternatively, we estimate $E(Y)$ by using Monte Carlo simulation:

$$E(Y) \approx \frac{1}{N}\sum_{n=1}^{N} y_n = \frac{1}{N}\sum_{n=1}^{N} \mathcal{T}_{\boldsymbol{\beta}_n}\Big( f\big(\boldsymbol{\theta}, \mathcal{T}_{\boldsymbol{\beta}_n}^{-1}(X - \boldsymbol{e}_n)\big)\Big), \text{where } \boldsymbol{\beta}_n \sim P_{\boldsymbol{\beta}}, \boldsymbol{e}_n \sim P_{\boldsymbol{e}} \quad (7)$$

where $N$ is the total number of simulation runs. In each simulation run, we first randomly sample $\boldsymbol{\beta}_n$ and $\boldsymbol{e}_n$ from the prior distributions $P_{\boldsymbol{\beta}}$ and $P_{\boldsymbol{e}}$, respectively. Then we obtain one possible hidden image with $\boldsymbol{\beta}_n$ and $\boldsymbol{e}_n$ based on Eq. (2), and feed it into the trained network to get its prediction, which is transformed with $\boldsymbol{\beta}_n$ to obtain $y_n$ according to Eq. (4). Therefore, this is an inference procedure with test-time augmentation.

## 2.3 Uncertainty estimation with test-time augmentation

The uncertainty is estimated by measuring how diverse the predictions for a given image are. Both the variance and entropy of the distribution $P(Y)$ can be used to estimate uncertainty. However, variance is not representative in the context of multi-modal distributions. In this paper we use entropy for uncertainty estimation.

$$H(Y) = -\int P(y)\ln\big(P(y)\big) dy \quad (8)$$

With the Monte Carlo simulation in Eq. (7), we can approximate $H(Y)$ from the simulation results $\mathcal{Y} = \{y_1, y_2, ..., y_N\}$. Suppose there are $M$ unique values in $\mathcal{Y}$ and the frequency of the $m$-th unique value is $\hat{p}_m$, then $H(Y)$ is approximated as:

$$H(Y) \approx -\sum_{m=1}^{M} \hat{p}_m \ln(\hat{p}_m) \quad (9)$$

For segmentation tasks, pixel-wise uncertainty estimation is desirable. Let $Y^i$ denote the predicted label for the $i$-th pixel. With the Monte Carlo simulation, a set of values for $Y^i$ are obtained $\mathcal{Y}^i = \{y_1^i, y_2^i, ..., y_N^i\}$. The entropy of the distribution of $Y^i$ is therefore approximated as:

$$H(Y^i) \approx -\sum_{m=1}^{M} \hat{p}_m^i \ln(\hat{p}_m^i) \quad (10)$$

where $\hat{p}_m^i$ is the frequency of the $m$-th unique value in $\mathcal{Y}^i$.

# 3 Experiments

We validated our proposed testing and uncertainty estimation method with two segmentation tasks: 2D fetal brain segmentation from MRI and 3D brain tumor segmentation from multi-modal MRI. In both tasks, we show how the inference with test-time augmentation affects segmentation accuracy, and analyze the uncertainty of the segmentation results. For a given trained CNN model, we compared four ways of testing: 1) the proposed test-time augmentation (TTA) for prediction, 2) test-time dropout (TTD) [2] where the output is an ensemble of $N$ predictions with random dropout at test time, 3) a combination of TTA and TTD where both TTA and TTD are used in all the testing runs, and 4) a single-prediction baseline that obtains the prediction without TTA and TTD. For the first three methods, the uncertainty was obtained by Eq. (10) with $N$ predictions. For TTD and TTA + TTD, the dropout probability was set as a typical value of 0.5. Prediction error rate and the Dice score between a segmentation result and the ground truth were used for quantitative measurements of segmentation accuracy.

## 3.1 2D fetal brain segmentation from MRI

### 3.1.1 Data and implementation

We collected clinical T2-weighted MRI scans of 60 fetuses in the second trimester with Single-Shot Fast Spin Echo (SSFSE). The data for each fetus contained three stacks of 2D slices acquired in axial,

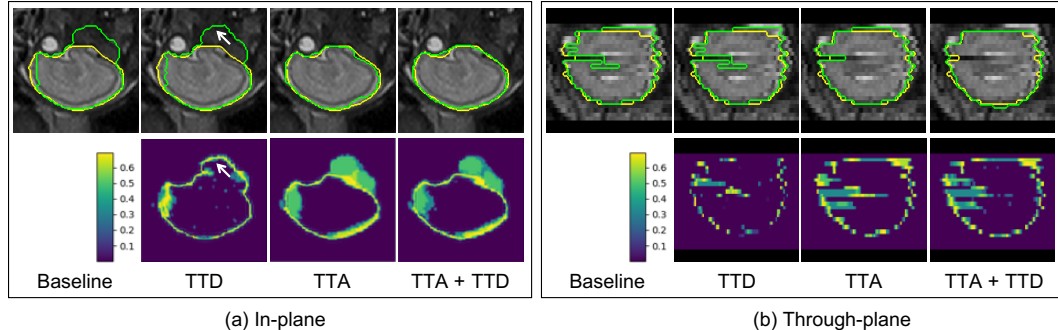

| Baseline | TTD | TTA | TTA + TTD | | Baseline | TTD | TTA | TTA + TTD |

(a) In-plane                              (b) Through-plane

Figure 1: Visual comparison of different testing methods for 2D segmentation of the fetal brain. First row: segmentation results (green curve) and ground truth (yellow curve). Second row: uncertainty maps with color bars. The white arrows show an area with high certainty while mis-segmented. TTD: test-time dropout, TTA: test-time augmentation.

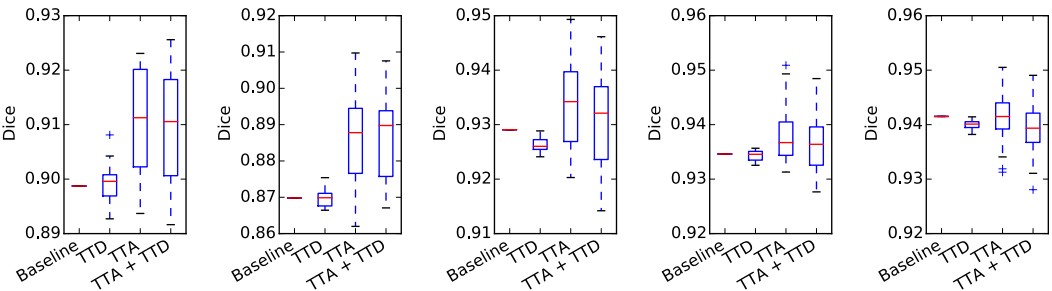

Figure 2: Dice distributions of segmentation results with different testing methods for five example stacks of 2D slices of fetal brain MRI.

sagittal and coronal views respectively, with pixel size 0.63 mm to 1.58 mm and slice thickness 3 mm to 6 mm. The gestational age ranged from 19 weeks to 33 weeks. We used 120 stacks of 40 patients for training, 12 stacks of 4 patients for validation and 48 stacks of 16 patients for testing. Manual segmentation results of these images were used as the ground truth. We normalized each stack by its intensity mean and standard deviation, and resampled each slice with pixel size 1.0 mm.

We used 2D networks of Fully Convolutional Network (FCN) [9], U-Net [13] and P-Net [15]. The networks were implemented in TensorFlow using NiftyNet [3]. During training, we used Adaptive Moment Estimation (Adam) to adjust the learning rate that was initialized as $10^{-3}$, with batch size 5, weight decay $10^{-7}$ and iteration number $10k$. We augmented the data by flipping along each axis with a probability of 0.5, rotation with an angle $r \sim U(0, 2\pi)$, scaling with a factor $s \sim U(0.8, 1.2)$, and adding random noise with $e \sim \mathcal{N}(0, 0.05)$, as a median-filter smoothed version of a normalized image in our dataset has a standard deviation around 0.95.

### 3.1.2 Segmentation results with uncertainty

Fig. 1 shows a visual comparison of four different testing methods for a fetal brain image. The results were based on the same trained model of U-Net, and the Monte Carlo simulation number $N$ was 20 for TTD, TTA, and TTA + TTD. The first row presents segmentation results compared with the ground truth. It shows the baseline obtained an obviously mis-segmented region outside the brain in Fig. 1(a), and the difference between the baseline and TTD is hardly observable. In comparison, TTA achieved better results than TTD. The difference between TTA and TTA + TTD is tiny. The second row presents the uncertainty maps for the segmentation results. For TTD, most of the uncertain segmentations are located near the border of the segmented foreground, while the pixels with a larger distance to the border have a very high confidence. This leads to a lot of overconfident incorrect segmentations, as shown by the white arrows in Fig. 1(a). In comparison, TTA obtained a larger uncertain area that is mainly corresponding to mis-segmented regions of the baseline. The uncertainty maps of TTA + TTD look similar to that of TTA.

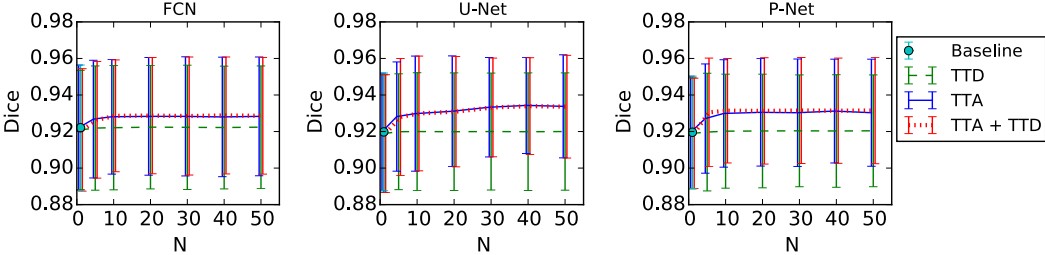

Figure 3: Dice of 2D fetal brain segmentation with the change of Monte Carlo simulation runs $N$.

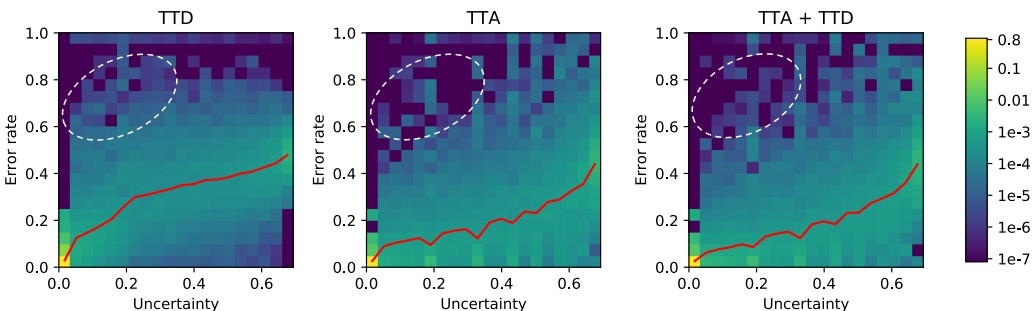

Figure 4: Normalized joint histogram of prediction uncertainty and error rate for 2D fetal brain segmentation. The average error rates at different uncertainty levels are depicted by the red curves. The dashed ellipses show that TTA reduces the occurrence of overconfident incorrect predictions.

To quantitatively evaluate the segmentation results, we measured Dice score of each prediction for different testing methods. Fig. 2 shows examples of five stacks of fetal brain MRI. The results were based on the same trained model of U-Net. Note that the baseline had only one prediction for each image, and the Monte Carlo simulation number $N$ was 20 for TTD, TTA and TTA + TTD. It can be observed that for each case, the Dice of TTD distributes nearly to that of the baseline. In comparison, the Dice distribution of TTA has a higher average and larger variance, which shows that TTA outperforms TTD in improving segmentation accuracy. Fig. 2 also shows that the performance of TTA + TTD is close to that of TTA.

We also measured the performance of different network structures with FCN [9], U-Net [13] and P-Net [15], and investigated how the segmentation accuracy changes with the increase of the Monte Carlo simulation runs $N$. The results measured with all the testing images are shown in Fig. 3. We found that for all of these three networks, the segmentation accuracy of TTD remains close to that of the baseline. For TTA and TTA + TTD, an improvement of segmentation accuracy can be observed when $N$ increases from 1 to 10. When $N$ is larger than 20, the segmentation accuracy for these two methods reaches a plateau.

To study the correlation between prediction uncertainty and accuracy, we measured the joint histograms of uncertainty and error rate for TTD, TTA, and TTA + TTD. Each histogram was obtained by statistically calculating the error rate of pixels at different uncertainty levels in each slice. The results based on U-Net with $N = 20$ are shown in Fig. 4, where the joint histograms have been normalized by the number of total pixels in the testing images for visualization. We calculated the average error rate at each uncertainty level, leading to a curve of error rate as a function of uncertainty, i.e., the red curves in Fig. 4. This figure shows that the majority of pixels have a low uncertainty with a small error rate. When the uncertainty increases, the error rate also improves gradually. However, when the prediction uncertainty is low, TTD has a steeper increase of error than TTA, which demonstrates that TTA has fewer overconfident incorrect predictions. The dashed ellipses in Fig. 4 also show the different levels of overconfident incorrect predictions for these testing methods.

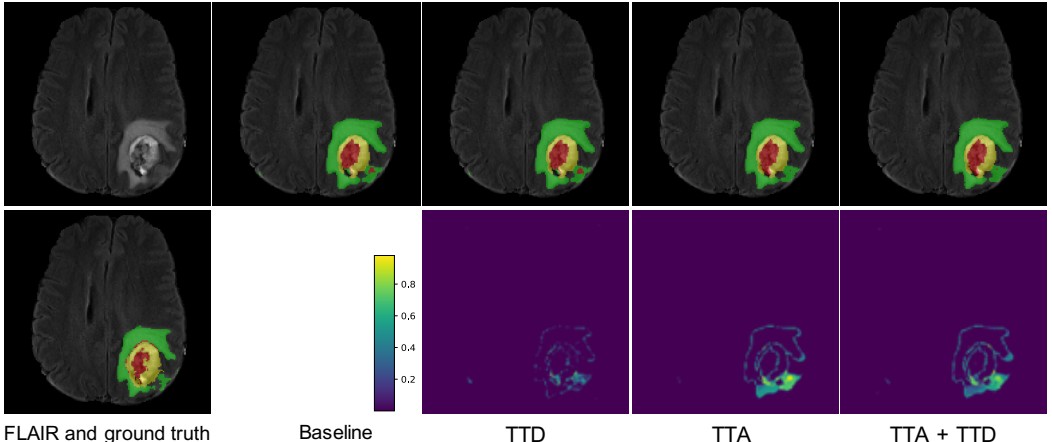

| FLAIR and ground truth | Baseline | TTD | TTA | TTA + TTD |

Figure 5: Visual comparison of different testing methods for 3D brain tumor segmentation. First row: FLAIR image and segmentation results (green: edema, yellow: enhancing core, red: necrotic core). Second row: ground truth and uncertainty maps.

### 3.2 3D brain tumor segmentation from multi-modal MRI

#### 3.2.1 Data and implementation

We used the BraTS 2017[3] [14] training dataset that consisted of volumetric images from 285 studies, with ground truth provided by the organizers. We randomly selected 20 studies for validation and 50 studies for testing, and used the remaining for training. For each study, there were four scans of T1-weighted, contrast enhanced T1-weighted (T1c), T2-weighted and Fluid Attenuation Inversion Recovery (FLAIR) images, and they had been co-registered. All the images were skull-stripped and re-sampled to an isotropic 1 mm$^3$ resolution. The task was to segment these multi-modal images into multiple classes: edema, enhancing core, necrotic core and the background (Fig. 5). We used 3D U-Net [1] and V-Net [11] implemented with NiftyNet [3], and employed Adam during training with initial learning rate $10^{-3}$, batch size 2, weight decay $10^{-7}$ and iteration number $20k$. We augmented the data by flipping along each axis with a probability of 0.5, rotation with an angle along each axis $r \sim U(0, 2\pi)$, scaling with a factor $s \sim U(0.8, 1.2)$, and adding random noise with $e \sim \mathcal{N}(0, 0.05)$ based on the reduced standard deviation of a median-filtered version of a normalized image.

#### 3.2.2 Segmentation results with uncertainty

Fig. 5 shows an example of segmentation results with different testing methods that used the same trained model of 3D U-Net. The Monte Carlo simulation number $N$ was 40 for TTD, TTA, and TTA + TTD. It can be observed that the baseline method led to an over segmentation of the necrotic core in the edema region. Compared with TTD, TTA and TTA + TTD had a better ability to correct this mis-segmentation. Fig. 5 also shows that TTA and TTA + TTD obtained their segmentations with higher uncertainties than TTD, especially in the region that was mis-segmented by the baseline.

We measured the error rate in each testing image at different uncertainty levels, and obtained the normalized joint histogram of prediction uncertainty and error rate. Fig. 6 shows the results based on 3D U-Net with $N = 40$. The red curve shows the average error rate as a function of prediction uncertainty. Fig. 6 shows that the average error rate increases with the growth of uncertainty. However, TTD has a higher average error rate than TTA and TTA + TTD when the uncertainty is low ($< 0.6$).

Following typical evaluation methods used for the BraTS dataset, we calculated the Dice scores for three structures: 1) the whole tumor including edema, enhancing core and necrotic core, 2) the tumor core without edema, and 3) enhancing tumor core [14]. We found that for 3D U-Net and V-Net, the performance of the multi-prediction testing methods reaches a plateau when $N$ is larger than 40. Table 1 shows the evaluation results with $N = 40$. It can be observed that for both networks,

---

[3]`http://www.med.upenn.edu/sbia/brats2017.html`

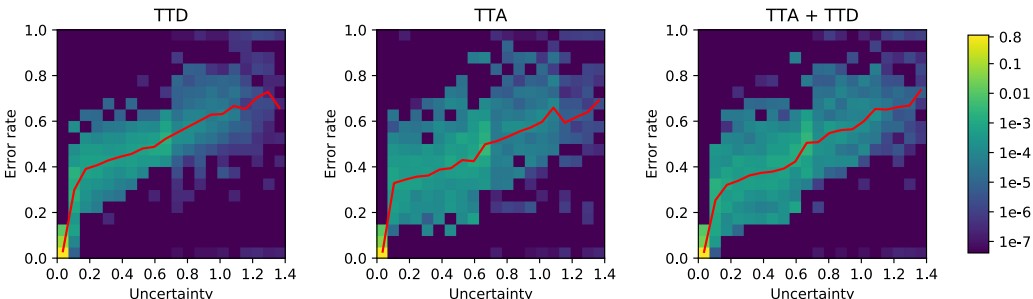

Figure 6: Normalized joint histogram of prediction uncertainty and error rate for 3D brain tumor segmentation. The average error rates at different uncertainty levels are depicted by the red curves.

Table 1: Dice (%) of 3D brain tumor segmentation by 3D U-Net [1] and V-Net [11] with different testing methods. WT: whole tumor, TC: tumor core, EC: enhancing core.

| | 3D U-Net | | | V-Net | | |
|---|---|---|---|---|---|---|
| | WT | TC | EC | WT | TC | EC |
| Baseline | 87.69±5.65 | 78.72±17.96 | 74.49±20.35 | 86.92±6.90 | 76.38±19.15 | 74.17±26.04 |
| TTD | 88.22±5.87 | 79.25±17.90 | 75.75±21.03 | 87.04±6.92 | 76.61±19.27 | 74.29±26.01 |
| TTA | 88.39±5.74 | 79.54±17.11 | **75.94±20.64** | 87.52±6.36 | **76.93±19.37** | 74.55±26.03 |
| TTA + TTD | **88.52±5.95** | **79.61±17.02** | 75.70±20.41 | **87.60±6.25** | 76.86±19.26 | **74.69±25.98** |

multi-prediction methods lead to better performance than the baseline with a single prediction, and TTA outperforms TTD in terms of average Dice score for all the three structures.

## 4   Discussion and conclusion

In our mathematical formulation of test-time augmentation based on an image acquisition model, we explicitly modeled spatial transformations and image noise. However, it can be easily extended to include more general transformations such as elastic deformations [1] or add a simulated bias field for MRI. In addition to the variation of possible values of model parameters, the prediction result is also dependent on the input data, e.g., image noise and transformations related to the object. Therefore, a good uncertainty estimation should take these factors into consideration. Fig. 1 and 5 show that model uncertainty alone is likely to obtain overconfident incorrect predictions, and TTA plays an important role in reducing such predictions. We have demonstrated TTA based on the image acquisition model for image segmentation tasks, but it is general for different image recognition tasks, such as image classification, object detection, and regression. For regression tasks where the outputs are not discretized category labels, the variation of the output distribution might be more suitable than entropy for uncertainty estimation.

In conclusion, we proposed a theoretical and mathematical formulation of test-time augmentation for medical image segmentation. With the formulation, we obtain the prediction by estimating its expectation with Monte Carlo simulation and modeling prior distributions of parameters in an image acquisition model. The formulation also leads to transformation-based uncertainty in the prediction. Experiments showed that TTA based on our formulation leads to higher segmentation accuracy than a single-prediction baseline and dropout-based multiple predictions, and demonstrated that our uncertainty estimation with TTA helps to reduce overconfident incorrect predictions encountered by model-based uncertainty estimation.

**Acknowledgments**

This work was supported through an Innovative Engineering for Health award by the Wellcome Trust (WT101957); Engineering and Physical Sciences Research Council (EPSRC) (NS/A000027/1, EP/H046410/1, EP/J020990/1, EP/K005278), Wellcome/EPSRC [203145Z/16/Z], the National

Institute for Health Research University College London Hospitals Biomedical Research Centre (NIHR BRC UCLH/UCL), the Royal Society [RG160569], and hardware donated by NVIDIA.

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
