# OpenReview forum: "Test-time augmentation with uncertainty estimation for deep learning-based medical image segmentation"
_MIDL.amsterdam/2018/Conference — Submitted to MIDL 2018_

### Review · AnonReviewer3 · 2018-05-07
**A paper that is well-written and easy to follow, but its significance requires further discussion**

**Rating:** 2
**Confidence:** 2

**Review:**

In this paper, the authors present a way to reduce false positives (here addressed as overconfident incorrect decisions) and thus improving deep learning model accuracy by introducing a test-time augmentation method. The paper is very well written and the logic easy to follow. However, I consider the following aspects shortcomings of this paper:
The results are presented nicely in this paper, and it is good to see the comparisons of different networks. Even though the uncertainty of the results are somewhat reduced, the accuracy improvement brought by the proposed method is very limited and insignificant.
The test time augmentation method of improving test accuracy in medical image processing is a very interesting idea. However, the authors’ proposed method of manipulating the test image by geometric transformations and image noises seem to lack the theoretical backup to me. As first of all, just like the authors wrote in the paper, it is a very common way to artificially enlarge the training data by geometrically transform the input training images, by doing this, the model accuracy can be improved slightly and the models would be more robust to rotations. Implementing this at the training time is very easy and takes minimum effort. Since the models built this way are already relatively robust to geometric transformations such as rotation, it is repetitive to do geometric transformations and compute the uncertainty at test time.
I personally do not see the reasons to manipulate the noises on the test image and compute uncertainty from those outputs. For that the noisy images do not represent the normal patterns of the classes trained on the training images, and using those results for uncertainty reduction and “improvements” of accuracy does not really make much sense to me.


**Special Issue:**

Yes

---

### Review · AnonReviewer2 · 2018-05-10
**Nice idea but no performance improvement and small evaluation data**

**Rating:** 2
**Confidence:** 2

**Review:**

The authors present a method that predicts an ensemble of predictions for multiple transformed versions of the input data. These transformations are sampled from specified prior distributions on noise and rigid image transformations. The method is evaluated on 48 stacks from 16 patients for testing. The authors compare their proposed, test-time augmentation method (TTA), against test-time dropout (TTD).

While I overall appreciate the idea of the paper to model uncertainty in the image data besides uncertainty in the model, I have concerns with the usability and evaluation of the proposed method.

* Based on the reported results, it seems that there is no statistical significant difference between the proposed method and the baseline.

* How much does TTA improved compared to a method that uses the proposed augmentations during training?

* It remains unclear what the impact of the weight initialization or the stopping point of training are.

* How should the hyper-parameters of the prior distributions be selected for a new application?

* How well are the uncertainty scores calibrated (see https://arxiv.org/pdf/1706.04599.pdf)?

* TTD seems to be a weak baseline. Methods like [7] have already reported stronger results and would be a more appropriate baseline. Do the authors intend to evaluate against other model-based uncertainty methods?

* The evaluation dataset of 48 stacks from 16 patients seems too small to draw strong conclusions from the presented accuracy results.




**Special Issue:**

No

---

### Review · AnonReviewer1 · 2018-05-12
**Good topic to delve into, but we do not learn too much from this study**

**Rating:** 2
**Confidence:** 2

**Review:**

This paper investigates a heuristic trick that has been used a lot in deep learning system design known as test time augmentation.

Pro:
- Interesting to analyze this aspect/trick of optimizing the performance of deep networks (and possibly other methods too) in more detail
- Well written

Comments:
- The fetal experiments use 2D networks for a 3D segmentation task and the results show through plane artifacts. This makes the results hardly relevant. There are obvious ways to improve the results here, so why focus on TTA?
- The literature review seems outdated. TTA has been used a lot, many papers mention and use it but do not refer to it in the abstract or title. Like ensembling, it seems to be a standard trick to eke out a bit more performance at the expense of some extra computation at test time. The authors describe it as some rarely used technique, but I think after http://benanne.github.io/2015/03/17/plankton.html it has been a standard component in the DL engineer's toolkit.
- I expect there must be some relationship between the use of augmentations during training and the added value of TTA. If a model is trained with a lot of augmentations it should start to give very similar output for augmentation at test time, and TTA should not add much. It would be interesting to investigate this in a paper like this.
- The differences in Table 1 are very small, are they significant?

**Special Issue:**

No

---

### Decision · Program_Chairs · 2018-05-15
**Paper51 Acceptance Decision**

Reject